# Investigation of Congo Red Toxicity towards Different Living Organisms: A Review

**Sharf Ilahi Siddiqui [1,2], Esam S. Allehyani [3], Sami A. Al-Harbi [3], Ziaul Hasan [4], May Abdullah Abomuti [5], Hament Kumar Rajor [1,*] and Seungdae Oh [6,*]**

[1] Department of Chemistry, Ramjas College, University of Delhi, New Delhi 110007, India; sharf_9793@rediff.com
[2] Department of Chemistry, Jamia Millia Islamia, New Delhi 110025, India
[3] Department of Chemistry, University College in Al-Jamoum, Umm Al-Qura University, Makkah 24382, Saudi Arabia
[4] Department of Biosciences, Jamia Millia Islamia, New Delhi 110025, India
[5] Department of Chemistry, Faculty of Science and Humanities, Shaqra University, Dawadmi 17472, Saudi Arabia
[6] Department of Civil Engineering, College of Engineering, Kyung Hee University, Yongin-si 17104, Gyeonggi-do, Republic of Korea
* Correspondence: rajorh@gmail.com (H.K.R.); soh@khu.ac.kr (S.O.)

**Abstract:** The use of dyes is widespread across almost all industries. Consequently, these dyes are found in various sources of water and food that humans, animals, and plants consume directly or indirectly. Most of these dyes are comprised of complex aromatic structures that have proven harmful. Congo red dye, a complex aromatic azo dye based on benzidine, is most commonly used in these dyes; its metabolites (benzidine and analogs) can be toxic, but Congo red dye itself is not always harmful. The present review summarizes the toxicity of Congo red dye towards different living forms. Herein, the primary emphasis has been given to the mutagenic, teratogenic, and carcinogenic consequences of Congo red and its metabolites. The mechanisms of azo dyes' carcinogenicity have also been discussed. This review will undoubtedly be beneficial for researchers to understand the harmful effects of Congo red in genotoxic, teratogenic, mutagenic, and carcinogenic factors.

**Keywords:** dyes; Congo red; benzidine; toxicity; reduction

## 1. Introduction

A dye is known for its coloring effects (a good factor) and its toxicity (a bad factor) [1]. Properties such as mutagenic, carcinogenic, and teratogenic define how toxic a dye is in nature [2]. The evaluation of these properties for a particular dye helps approve the dye's production and use. The paper and textile industries run vastly throughout the world. They are the source of employment for many workers and produce colored compounds for coloring materials. These coloring compounds are applied to the materials to enhance aesthetic value [3].

As a coloring compound, azo dyes are the most suitable option. These are a crucial class of dyes, responsible for exceeding 50% of yearly global manufacturing and about two-thirds of all synthetic dyes [4–6]. Azo dyes may be water-soluble or water-insoluble and are considered toxic, mutagenic, and carcinogenic agents. The carcinogen may be a dye itself or its metabolites. In water-soluble dyes, the metabolite is the carcinogen readily absorbed by the body [7]. The adverse effects of azo dyes on the ecosystem account for a longer duration. The complex aromatic structure of dyes, which depend on aromatic rings, is a major factor in their toxicity [8]. The acute toxicity of azo dyes is determined to be very low, with 50% lethal dose ($LD_{50}$) values down to 250–2000 mg kg$^{-1}$ body weight, as mentioned by European union guidelines for the classification of hazardous compounds [9].

## 2. Congo Red

Congo-red (CR) is an azo dye with the molecular formula of $C_{32}H_{22}N_6Na_2O_6S_2$ and molecular weight = 696.68 g mol$^{-1}$. Paul Bottinger discovered CR as the first direct dye in 1884 [10]. It is an anionic di-azo dye (contains two groups -N=N-) composed of a sodium salt of benzidinediazo-bis-1-naphthylamine-4-sulfonic acid, known by common names such as CR 4B, C.I. 22120, Cotton red B, Cotton red C, Direct red 28, Cosmos red, Direct red Y, and Direct red R [11–14].

CR contains two azo (-N=N-) chromophores and acidic auxochromes (-SO$_3$H) linked with the benzene structures [12,13]. CR is also called acidic diazo dye. A molecule of CR is linearly symmetrical, with a hydrophobic center consisting of two phenyl rings connected via di-azo linkage [14]. The phenyl rings are connected to two charged terminal naphthalene moieties containing sulfonic and amino groups [15]. The chemical structure of CR is shown in Figure 1, and it is made by mixing two molecules of naphthenic acid with tetrazotized benzidine [16]. A blue dye is obtained and converted into red disodium salt while salting out with sodium chloride.

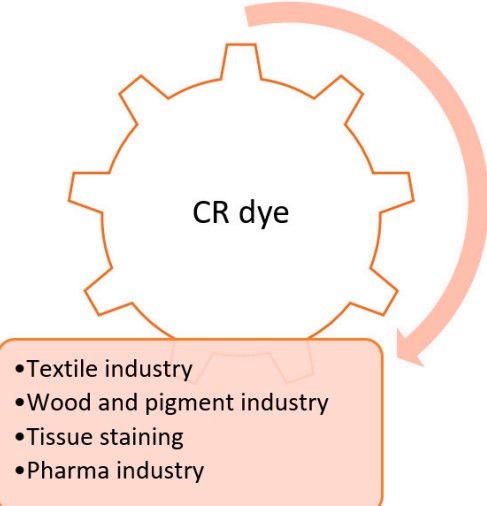

**Figure 1.** Molecular structure of CR. Adapted from the literature [16].

CR's color index number ranges from 2000 to 29,999. CR is also called acidic di-azo dye and is water soluble. CR gives a characteristic color in an aqueous solution due to the presence of an azo structure. It appears red in the basic medium and blue in the acidic medium because it is a di-azo dye. CR can form an amine constituent such as benzidine on the cleavage of its azo groups. Benzidine is a common carcinogen, so CR is included under the category of banned azo dyes [14]. Although CR finds practical applications in many industries like textile, cosmetics, pigment, leather, food, pharmaceutical, pulp, and paper, its extensive use also leads to industrial pollution. Figure 2 illustrates the major sources of CR pollution in the environment [17].

CR dye

- Textile industry
- Wood and pigment industry
- Tissue staining
- Pharma industry

**Figure 2.** Major sources of CR contamination in the environment.

## 3. Toxicities Caused by CR

Due to its widespread use, it is rather difficult to check and control the adverse effects of any material that may otherwise prove to be a potential toxicant. However, a wide variety of toxic effects have been observed with CR dye, as shown in Table 1. The following sections discuss the different life forms adversely affected by CR toxicity.

**Table 1.** Toxic effects showed by CR and the affected targets.

| Order | Toxic/Inhibitory Effect | Affected Targets | References |
|:-:|:-:|:-:|:-:|
| 1. | Carcinogenic | Humans and animals | [18] |
| 2. | Mutagenic | Humans and animals | [19] |
| 3. | Causing Infertility | Water flea (*Ceriodaphnia dubia*) | [20] |
| 4. | Increases COD | Water bodies and aquatic flora and fauna | [21] |
| 5. | It makes surface water unaesthetic | Water bodies | [22] |
| 6. | Allergic | Humans | [23] |
| 7. | Phytotoxicity | Plants | [24] |

### 3.1. Toxic Effect of CR on Humans

CR adversely affects the human body leading to various diseases, which can be fatal, when its concentration is high from being cytotoxic (genotoxic, hemotoxic, and neurotoxic), carcinogenic, and mutagenic [25]. Among the organs, it affects the eyes, skin, respiratory, and reproductive systems. Benzidine (its toxic metabolite) causes an allergic reaction and is a carcinogenic product. Benzidine is a bladder carcinogen and binds covalently to cellular macromolecules leading to activity inhibition. In animal experiments, benzidine-based dyes produce hepatocarcinoma, splenic sarcoma, nuclear abnormalities, and chromosomal errors in mammalian cells [26]. CR can cause platelet aggregation, thrombocytopenia, and disseminated micro-embolism by lowering blood protein content [15].

As reported earlier, CR has been known to cause an inhibitory effect on the activities of two enzymes: aspartate aminotransferase and alanine aminotransferase, mainly found in the liver. During tissue traumas such as cellular necrosis and cell growth, both enzymes are dispersed intracellularly and escape into the bloodstream. Occasionally, when tissue damage occurs, the activities of the enzymes are raised [27].

As reported, CR interacts covalently with proteins in organs other than the liver. One of the reasons could be that the function of enzymes is bound to be influenced since they are proteins. Benzidine, the metabolic product of CR, is mainly behind enzyme activity inhibition. Moreover, CR was also found to lower glutathione levels in rabbit and rat serum [27]. According to a study, CR in high dilution hastens the coagulation of cats' blood, whereas higher dye concentrations may slow clotting and retain the blood in a fluid state [28].

### 3.2. The Toxic Effect of CR on Aquatic Species

To be mutagenic, large proportions of azo dyes demand chemical activation, such as azo link reduction and cleavage to aromatic amines (AA). The biological toxicities of these dyes were seen for various pathogens species such as bacterium (*Vibrio fischeri*, *V. fischeri*), microalga (*Selenastrum capricornutum*, *S. capricornutum*), ciliate (*Tetrahymena pyriformis*, *T. pyriformis*) [29], Cladoceran (*Daphnia magna*, *D. magna* and *Ceriodaphnia rigaudi*, *C. rigaudi*), and Zebrafish (*Danio rerio*. *Pseudokirchneriella subcapitata*, *D. rerio*. *P. subcapitata*) [30]. The level of toxicity ranges from species to species.

Bacterial, algal, protozoan, Ames, and flash bioluminescence tests were employed to know the dyes' toxicity [29]. Toxicological tests supplement cytotoxicity data, including morphological and physiological assessments.

### 3.2.1. Bacterial Test

CR dye is known to be carcinogenic and harms the intestinal bacteria involved in digesting it [31]. Apart from the positive contribution of intestinal bacteria to our health, it can also pose a negative effect. The chemical reduction and breaking of azo bonds by *azoreductase*, which is found in bacteria, cause the toxicity of azo dye. *Rhodococcus rhodochrus (R. rhodochrus)* is one of the bacteria that make up our intestinal microbiota, capable of reduction and cleavage of a large number of organic compounds [31,32]. The aerobic Gram-positive bacteria can also metabolize CR. Despite the *R. rhodochrous* metabolites inserted into the bacterial culture, they cause a toxic effect on bacterial growth. The higher the dye metabolized, the higher the toxicity of the dye to bacterial growth [31].

### 3.2.2. Algal Test

The algal test is considered to be the most sensitive to dyes compared with other tests. Nevertheless, CR toxicity can be diagnosed with a variety of biochemical tests involving the zebrafish (*D. rerio*), the microalga (*P. subcapitata*), and the cladocerans (*D. magna* and *C. rigaudi*).

*P. subcapitata* was the most susceptible to CR, with half maximal inhibitory concentration ($IC_{50}$) being 3.11 mg/L [30]. *P. subcapitata* growth was affected at all concentration ranges. With increasing CR concentrations, the number of photosynthetic pigments, chlorophyll A and β, carotenoid content per cell increased. The impact of CR on macromolecule content (proteins, carbs, and fats) was also reported to be increased. The cladocerans (*D. magna* and *C. rigaudi*) survival decreased as CR concentration increased. *D. magna*, on the other hand, was more resistant to CR dye than *C. dubia*. Several consequences of azo dyes were observed in zebrafish embryos, ranging from malformations to hatching failures. Zebrafish embryos are not acutely harmed by CR dye; it produced sublethal effects. Among those effects were edema in the yolk sac, skeletal abnormalities, and delayed or precluded hatching. Hernández-Zamora and Martínez-Jerónimo [30] reported the reduction in the growth rate, photosynthesis, and respiration that CR has caused in green microalgae species *Chlorella vulgaris* (*C. vulgaris*). The growth rate is reduced due to a decrease in metabolic activity. The metabolic rate (photosynthesis and respiration) was suppressed at all CR concentrations. The photosynthetic and respiratory processes were reduced by 84 and 98 percent and 76 and 96 percent, respectively, at 5 and 25 mg/L concentrations. The cells continue to develop slower even when the photosynthetic and respiratory rates drop dramatically. Photosystem II (PSII) changes were found in response to CR at 10 to 25 mg/L. The donor site of PSII is notably affected in terms of photosynthetic activity. Non-photochemical thermal dissipation pathways increase due to the decreased ability to absorb and utilize quantum energy.

### 3.2.3. Flash Bioluminescence Test

A flash bioluminescence test was used to determine the acute toxicity of the dark-colored compound. However, the sensitivity of the flash bioluminescence test was found to be low for azo dyes such as CR. The $EC_{50}$ value of CR was found to be 1623 mg/L with *V. fischeri* bacterium [29].

### 3.2.4. Ames Test

The Ames test measured the dyes' genetic toxicity with and without metabolic activation. It was found to be more sensitive than the flash bioluminescence test. The $EC_{50}$ value of CR was found to be $4.8 \pm 1.0$ mg/L with microalga, *S. capricornutum* [29].

### 3.2.5. Protozoan Tests

Protozoan tests were also used to assess the toxicological potential of azo dyes in aquatic environments using ciliated protist *T. pyriformis*. A dose of 500 mg/L of dyes was used. The test typically included a grazing experiment and morphometric analysis (cell area and cell width/length (W/L) ratio), as well as a protocol developed for measuring

population growth impairment and generation time [29]. After a 24-h exposure duration, CR was observed to modify the cell area value [29] drastically.

### 3.2.6. Genetic Toxicity Test

To detect substitution mutations and shift mutations, auxotrophic strains *Salmonella typhimurium* (*S. typhimurium*), *His* TA100 and YG1042 strains, and TA98 and YG1041 were used, respectively. The dye concentration per agar plate ranges from 50 to 400 μg. Including sulfonic groups in the dye molecule, which can reduce the mutagenic effect, may explain the negative results reported with Remazol Brilliant Blue R and CR [29].

### 3.3. The Toxic Effect of CR on Plants

The severity of azo dye's toxicity (genotoxic and cytotoxic) is also visible in aquatic plant life. Azo dye, when mixed with water, reduces the penetration of light which alters the photosynthesis process, and, thus, the ecosystem is negatively affected [33]. When released into water, it not only raises the pH, biological oxygen demand (BOD), and chemical oxygen demand (COD) of the water but also disrupts the environment's organic-inorganic chemical equilibrium. The biotic content of water is affected by these organic-inorganic chemical concentrations in the environment [34].

As reported in earlier literature [35], CR is proven to be toxic for the aquatic plant species, *Lemna minor* (LM), point of view physiological and cytogenetic. According to reports, altering the CR concentrations has several negative consequences on the LM plant including root growth, fresh mass at 5 parts per million (ppm), and reduced total frond surface. Above 2500 ppm, CR inhibits LM plant growth, decreases chlorophyll A content, and increases carotenoid content. Above 1000 ppm, it also shows a considerable decline in PSII efficiency, a fall in mitotic indices at 5 and 1000 ppm dye, and zero at 5000 ppm. Additionally, at 5 and 1000 ppm dye, an increase in the figure of chromosomal abnormalities was observed, and a 56 percent decontamination of the growing medium has been reported at 250 ppm. Another study reported CR toxic effects on the LM plant [36]. At all CR values more than 1000 ppm, this study found an increase in chlorophyll B. LM plants underwent certain stress by CR due to elevated carotenoid amounts and reduced chlorophyll fluorescence values. This supposition is further sustained by the higher phenolic contents in treated plants, as it is known that such parameters may reflect abiotic stress levels. Two other azo dyes (acid scarlet GR or acid red B) have been found to have adverse effects on two plant species, *Medicago sativa L* (*M. sativa L*) and *Sesbania cannabina Pers* (*S. cannabina pers*). These plant species were resistant to azo dyes at low concentrations. Germination rate and root elongation were not considerably reduced in *Medicago sativa L.* plants at concentrations below 1 g/L, but significantly reduced at 5 g/L. Even though the azo dyes were used at modest concentrations, root elongation in *S. cannabina pers* was reduced [37].

### 3.4. Evaluation of Toxic, Mutagenic, Carcinogenic, and Teratogenic Properties

A dye or a substance is said to possess toxicity if it starts damaging the living organism at a particular dose. Toxicity can be acute (even in a minor or one-time exposure) or chronic (a measure of how toxic it is over a more extended period may be from weeks to years) [38]. A substance is said to be carcinogenic if it causes or promotes the occurrence of cancer. The mutagenic effect is known for causing an irreversible alteration in the quantity or anatomy of the genetic material in a cell [39]. A safe dye for the environment can be evaluated based on its genotoxicity, carcinogenic, and mutagenic effects.

Dye reductively cleaves after oral ingestion or when it comes into contact with saliva, sweat, or gastric fluids and forms AA (diphenylamine such as benzidine and 4-diphenylamine) [40]. To a lesser extent, the reduction of azo dyes can be accomplished by human skin microflora, intestinal microflora, ambient microorganisms, human liver azoreductase in the intestinal wall or liver, and non-biological means [41].

Another way azo dyes are degraded is through the cleavage of aromatic R–NH$_2$ amines. Photodegradation and biodegradation via hydroxylation, oxidation, and hydrolysis are the other methods of degradation. Naseem et al. [25] reported a reduction mechanism of CR reduction by using NaBH$_4$. When dissolved in aqueous media, CR was not reduced by adding reducing agents; some electrons are needed. It requires a surface (nanoparticles) to attach the CR molecules with a reducing agent where binding and product formation occurs. During CR reduction, surface-providing metal nanoparticles operate as a transfer point, transporting electrons from BH$_4^{-1}$ ions to CR molecules by acting as electron donors and CR molecules as electron acceptors. The azo bonds are broken during the reduction of CR, which causes it to turn colorless and produce AA [25].

On the other hand, the terminal amine in organisms is due to the biological degradation of the azo dye in humans [42].

A high proportion of mammalian species, which includes mice [43], rhesus monkeys [44], and humans [7], have been observed to reduce orally consumed azo compounds to AA. Dyestuffs provide only a tiny portion of the mutagenesis potential. Azo dyes make up around 80% of the dyestuffs used in textile dyeing. Under reducing conditions, several of these dyes may cleave off, releasing AA that is carcinogenic in animal studies. Azo dyes need metabolic activation, such as azo bond reduction and cleavage; otherwise, they will be photochemically azo-reduced to the equivalent AA components, resulting in mutagenicity [29,45]. Mutagenicity and carcinogenicity are caused by azo dyes and their metabolites (such as benzidine, arylamines, diamines, and free radicals) [46].

The carcinogenicity of azo dyes is intimately related to the molecule's structure and degradation mechanism. Because of their possible carcinogenicity, AA is one of the degradation products and pollutants of significant concern. The primary enzyme-catalyzed processes involved in the biotransformation of dyes include oxidation, reduction, hydrolysis, and conjugation [47]; however, biological reductions and oxidations of azo dyes are responsible for the possible existence of hazardous amines in humans [47,48]. After an azo dye is reduced an AA is produced, which is metabolically oxidized to reactive electrophilic species before binding to DNA in an irreversible process [34]. The intestinal microflora has an important role in activating azo dyes [49,50]. The CYP450 enzymes available in the intestine could potentially play a role in activating these dyes [40,41]. According to experimental findings, CR was found to have teratogenic qualities in addition to carcinogenic and mutagenic properties [51].

On the eighth day of gestation, pregnant rats were given an intraperitoneal injection of a 2 percent aqueous solution of CR at 20 mg per 100 g. As a result, CR causes fetuses to develop *hydronephrosis*, *hydrocephalus*, *microphthalmia*, or *anophthalmia*. A 40 mg per 100 g dosage of CR in a pregnant rat was proven fatal [51]. The CR LD$_{50}$ value was estimated to be 190 mg/ kg in rats [15].

### 3.5. Groups of CR That Cause Toxicity

#### 3.5.1. Aromatic Amines (AA)

AA are aromatic hydrocarbons containing amino substituents. They are potent mutagenic, carcinogenic, and hemotoxic agents absorbed from the skin, gut, and respiratory tract. The major events in its chemical carcinogenesis are metabolic stimulation and DNA adduction [52].

AA is easily metabolized by the enzyme system located in the liver. They yield metabolites that undergo redox cycling [53]. A metabolic attack on AA is usually oxidation: oxidation of N atom (N-oxidation) and oxidation of carbon of aromatic ring (C-oxidation). Amine hydroxylation and subsequent esterification are key initial steps in the metabolic activation of arylamines. The aromatic ring's metabolic hydroxylation occurs when the AA's free amine group is activated (C-hydroxylation). AA metabolites thus produced constitute the natural carcinogenic agents.

Conjugation of the metabolite of AA with glucuronic acid is the most important detoxification process. AA given to rats is eliminated as ethereal sulfates or amino phenol

glucuronides. 4-amino diphenyl, benzidine, 4-amino stilbene, and 4-acetylamino stilbene are "two benzene ring" amines excreted primarily as glucuronides. Glucuronides have now been proven to cause cancer [54]. AA is also known for initiating tumor formation by modifying tissue macromolecules. The conversion of hydroxamic acid derivatives to reactive, hazardous sulfate conjugates has been linked to the induction of liver tumors in rats caused by primary AA. Reactive metabolite generated via pathways involving peroxidation is another mechanism for activating the AA. In vivo AA reactivity is determined by metabolic activation. N-hydroxylation of the amine group is followed by sulfation and acetylation conjugation. The nitrenium ion is regarded as the ultimate reactive metabolite that interacts with and destroys DNA. Chronic exposure to AA in the workplace poses a significant risk of bladder cancer; several of them, such as 4-amino diphenyl, auramine, benzidine, -naphthylamine, O-dianisidine, and methylene bis O-chloroaniline are proven bladder carcinogens [55]. Some of the carcinogenic AA are present in the environment, accumulating in food chains and constituting a risk to human life and the environment in general [42,48].

### 3.5.2. Benzidine

Previous studies have reported that, due to intestinal microbiota, water-soluble azo dyes undergo metabolic reduction to produce dye subunits such as benzidine [8,56]. Benzidine is a member of a known carcinogen called AA, a biphenyl amine with two covalently linked benzene rings (1,1) that an amino group replaces at 4,4′. It is made by reducing nitrobenzenes and then rearranging the resultant hydrazobenzenes intramolecularly with an acid catalyst. In numerous species, including humans, dogs, mice, rats, and hamsters, benzidine causes bladder cancer; one of its notable side effects. Benzidine and its congeners have been found in the urine of humans and animals subjected to benzidine-based dyes [57]. In the cells of rodents in vivo, benzidine covalently binds to DNA, causing chromosomal damage, sister chromatid exchanges, micronuclei development, DNA strand breakage, and unscheduled DNA synthesis. Benzidine induces various human and animal tumors [26,56]. Most carcinogenic benzidine analogs are also mutagenic, and one of the fundamental factors in their carcinogenesis is their metabolism to electrophiles, which interact with DNA and produce mutations [58].

### 3.6. Some Major Findings Related to CR

CR is a nonmutagenic azo dye, but its metabolite benzidine is a known mutagen and has been linked to human urinary bladder cancer as discussed earlier [59,60]. According to Tanaka and colleagues, some human patients whose stomachs were sprayed with CR during a gastrointestinal endoscopy contained mutagenic urine [61]. This study shows that rat cecal bacteria can lower CR and release benzidine [59–61]. The amount of carcinogenic activity retrieved from the bacterial reduction incubation is also proportional to the amount of benzidine present in the dye at the testing time. Dye impurities, reduction products other than benzidine, and further benzidine metabolism by cecal bacteria appear to have little effect on the reduced dye's mutagenicity [62].

Rat cecal bacteria also cause a 95% drop in CR to its parent diamine, benzidine, according to the study. In a conventional Ames test, which uses rat liver S9 to metabolize benzidine to its ultimate mutagenic state (s), the extracts from these reduction incubations are mutagenic [62]. The aminobiphenyl group and azo linkages are two characteristics that distinguish CR as recalcitrant and xenobiotic. The metabolism of these benzidine-based dyes results in the release of free benzidine and the activation of chromosomal abnormalities in humans and animals [26]. A mammalian intestine bacterial system was used to evaluate CR's azo-reduction activity, which revealed that CR trims reductively to yield free benzidine [63].

In the *Salmonella* tester strain, CR induced its mutagenic activity before its azo reduction. For rat liver homogenate S9 activation, CR was not acting as a direct mutagen. The incubation of intestinal anaerobic bacteria was pre-step for inducing the activation. The

intestinal bacteria then produce corresponding benzidine constituents quantitatively after reduction. In the *Salmonella* experiment, pre-incubation caused mutagenicity in rat gut bacterial cells that were substantially equal to the same amount of free benzidine tested straightaway with S9 activation [62].

Martin and Kenelly [64] conducted a comparative investigation in anaerobic intestine bacterial systems for certain rat liver microsomal azo-reductase preparations to evaluate the reduction rates of non-benzidine azo dyes, many benzidines, and 3,3'-substituted benzidine azo dyes. Non-benzidine azo dyes were reported to have higher degradation rates with rat liver microsomes than benzidines and 3,3-substituted benzidine azo dyes. CR had a reduced rate of <2% of Amaranth dye. After reduction by anaerobic intestine bacterial systems, benzidines, and 3,3-substituted benzidine azo dyes appeared to be weak substrates for a few rats' liver microsomal azo-reductase formulations. Additionally, oxygen was shown not to affect this rat liver azo reduction formulation [64].

The covalent binding of CR to rat liver was discovered by Kenelly and others [64]. At seven days after dosing, the DNA of I.P. administered CR demonstrated carcinogenicity larger than fivefold. It effectively produced free benzidine in mammalian intestine anaerobic bacteria [63].

### 3.7. Mechanisms for Azo Dye Carcinogenicity

Azo dye toxicology primarily focuses on the hypothesis that azo linkage is cleaved upon reduction and forms the corresponding amines. This is an important metabolic step on the pathway to the dye's carcinogenic activation. Brown and Vito [45] reported three different azo dye carcinogenicity mechanisms involving metabolic activation to reactive electrophilic intermediates. An anaerobic bacterium in the intestine reduces azo dyes by cleaving the azo bond and forming AA. This AA covalently binds to DNA after its metabolic oxidation into reactive electrophilic species. Azo dyes with free AA groups can be metabolically oxidized without the necessity for azo reduction. Direct oxidation of azo linkage to highly reactive electrophilic diazonium salts may cause the activation of azo dyes [45].

### 3.8. Lethal Dose Low

The lowest dose of harmful material at which an exposed animal dies under controlled conditions is known as the lethal dose low (LDLo). The dosage is given per unit of body weight (usually reported in milligrams per kilogram) of a drug that has been linked to the death of a specific species. The administration methods are ingestion, inhalation, and intravenous [14]. CR toxicity for different species by knowing its LDLo value is given in Table 2.

**Table 2.** CR toxicity in the various organism by comparing the LDLo values. Adapted from the NCBI [14].

| Order | Organism | Test Type | Route | Dose | Effect |
|:-----:|:--------:|:---------:|:-----:|:-----|:------:|
| 1 | Man | Ldlo | Oral | 143 mg/kg | Vascular: other changes |
| 2 | Man | Ldlo | Intravenous | 1429 ug/kg | Behavioral: convulsions or effect on seizure threshold; lungs, thorax, or respiration: dyspnea; blood: change in clotting factors |
| 3 | Rat | Ldlo | Inhalation | 50 gm/m3/1H | - |
| 4 | Rat | Ldlo | Intravenous | 160 mg/kg | Behavioral: somnolence (general depressed activity); lungs, thorax, or respiration: chronic pulmonary edema; blood: change in clotting factors |
| 5 | Mouse | Ldlo | Intravenous | 250 mg/kg | |

**Table 2.** *Cont.*

| Order | Organism | Test Type | Route | Dose | Effect |
|:---:|:---:|:---:|:---:|:---:|:---:|
| 6 | Cat | Ldlo | Intravenous | 100 mg/kg | Behavioral: somnolence (general depressed activity); lungs, thorax, or respiration: chronic pulmonary edema; blood: change in clotting factors |
| 7 | Rabbit | Ldlo | Oral | 6 gm/kg | - |
| 8 | Rabbit | Ldlo | Skin | 4 gm/kg | - |
| 9 | Rabbit | Ldlo | Intravenous | 230 mg/kg | Behavioral: somnolence (general depressed activity); lungs, thorax, or respiration: chronic pulmonary edema; blood: change in clotting factors |
| 10 | Pigeon | Ldlo | Intravenous | 120 mg/kg | Behavioral: somnolence (general depressed activity); lungs, thorax, or respiration: chronic pulmonary edema; blood: change in clotting factors |

## 4. Conclusions

Restrictions on the use of azo dyes have been noticed in various countries. Although azo dyes add aesthetic value to the material, their toxicity poses a serious problem. As a result of their release into the water, these dyes cause water pollution and adversely affect different living forms and, as a result, our ecosystem. Consequently, their toxicity is a grave concern. CR is toxic to a wide variety of organisms, including humans, plants, animals, bacteria, algae, protozoa, and protozoa. Besides its carcinogenic properties, it is mutagenic, teratogenic, and teratogenic. Of note, CR itself is not toxic, but after its reduction to corresponding amines, it becomes toxic. We take preventive measures to safeguard different living forms to minimize the adverse effects and fatalities caused by various azo dyes.

**Author Contributions:** H.K.R., S.I.S. and S.O.: Formal Analysis, Investigation, Methodology, and Writing—original draft. E.S.A., S.A.A.-H., M.A.A. and Z.H.: Writing—review & editing. S.O.: Funding acquisition, supervision, validation, and writing—review & editing. All authors have read and agreed to the published version of the manuscript.

**Funding:** This study was supported by a National Research Foundation of Korea (NRF) grant funded by the Korean government (MSIP; Ministry of Science, ICT & Future Planning) (No. NRF-2020R1A2C110157311).

**Data Availability Statement:** All the data and materials related to the manuscript are published with the paper, and available from the corresponding author upon request.

**Acknowledgments:** This study was supported by a National Research Foundation of Korea (NRF) grant funded by the Korean government (MSIP; Ministry of Science, ICT & Future Planning) (No. NRF-2020R1A2C110157311).

**Conflicts of Interest:** The authors declare that they have no conflict of interests.

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
