# Peer review of "Investigation of Congo Red Toxicity towards Different Living Organisms: A Review"

_processes, doi:10.3390/pr11030807_

Round 1

Reviewer 1 Report

Dear Authors,

The strength of your manuscript is the attention to the mutagenetic, carcinogenetic, and toxicity of Congo red dye. The concerns of your manuscript are listed below.

General comments

1. “Congo red” or “Congo Red”. Which one of these are you going to choose? Recheck your manuscript very carefully (including the content, figures, and figure legends).

2. I found many errors in your writing, including font size, word spacing, capitalization, and italicizing errors. Recheck your manuscript very carefully.

3. For all scientific names, you should italicize them. Furthermore, it should “abbreviate the genera name(s)” as this is the second time the names have been used. Recheck your manuscript very carefully.

1. Introduction

1.1 Rewrite the sentence “Evaluation of these properties can help in the approval or ban of a particular dye.”. Please change the word “ban” to another word.

2. Congo red

2.1 Figure 1, Please change the word “water soluble-groups” to “Water soluble-groups”.

2.2 Please maximize the resolution of Figure 1.

2.3 Recheck the capitalization and font size errors.

2.4 Recheck the choice of the word “Congo red”.

3. Toxicities cause by CR

3.1 Table 1, Lowercase the word “Infertility”.

3.2 Table 1, Please inform the common name of Ceriodaphnia dubia.

3.3 Figure 3, Remove the pi-symbol from the word Congo red dye.

3.4 Figure 3, Please revise this figure. It needs to be better.

3.5 Section 3.2.2 line 155, Please change the word “PSII” to “Photosystem II (PSII)”.

3.6 Section 3.2.2 line 156, Please change the word “photosystem II” to “PSII”.

3.7 Section, 3.2.5, line 174, Revise the reference, Dias et al. (2003) to bracket format.  

3.8 Section 3.5, line 219, Please change the word “AA” to “aromatic amines (AA)”.

3.9 Section 3.6.1, line 284, Italicize the word “in vivo”.

3.10 Table 2, All references disappeared.

3.11 Table 2, table legend, “LDLO” or “LDLo”. Which one of these are you going to choose?

3.12 Recheck the scientific name errors, especially sections 3.2.1, 3.2.2, 3.2.6, 3.4, and 3.7.

3.13 Recheck the word spacing and font size errors.

4. Conclusions

4.1 Rewrite the sentence “Banned of various azo dyes have been noticed in some countries.”. Please change the word “banned” to another word.

5. Conflicts of Interest

5.1 Recheck the font size error.

Reviewer 2 Report

This manuscript describes a major problem of azo dyes contaminating the water. But this investigation is not well presented and the results that are given are classic and repetitive so this manuscript does not present any evidence.

Reviewer 3 Report

The authors bring in this article a relevant contribution about the toxicity of azo dyes to the literature on these chemical species, however there are still some minor issues that need to be corrected for the acceptance of the work, as described in annex.

Reviewer 4 Report

Reviewer

The manuscript entitled “Investigation of Congo red toxicity towards different living forms: A review” by Bushra Fatima et al. has been reviewed. I suggest its acceptance after the major revision.

 Comment 1

Manuscript presentation to be improved with the help of English/ communication expert.

Comment 2

Attention should be paid to abbreviations. For example: COD; S. No.

Comment 3

Congo-red or Congo red - It should be a uniform thought manuscript.

Comment 4

I would suggest that the title of the manuscript should be changed. For example: living organisms.

Comment 5

Please rewrite the abstract- it can be confusing for readers.

Comment 6 

References should be uniforms.

Comment 7

„Generally toxicity if dyes depends on the fact that how many benzene rings it had“ – please rewrite this sentance.

Comment 8 

line 51:  molecular formula and molecular weight

Comment 9

Common names - common names

Comment 10

Could you please display more clearly Figure 1, 2 and 3. All figures must be improved. Please add references for all figures and in Table 2.

Comment 11

auxochromes in Congo red- should be explained

Comment 12

chronical and acute toxicity to various live ogranisms should be improved with detailed information.

Comment 13

One such bacteria is Rhodococcus rhodochrous - Please rewrite this sentence.

Comment 14

milligrammes per kilogramme - milligrams per kilogram

Round 2

Reviewer 1 Report

After I reviewed this manuscript, I was satisfied with the revised version. Every reviewer's comment was responded to with clear answers. 

Author Response

Reply: Authors are very thankful to reviewer for his/her positive response and acceptance.   

Reviewer 2 Report

In general, this manuscript is well structured and respects journal standards. the references are very old, it is reinforced by recent references (last 3 years)

Author Response

Reply: We have replaced the old reference with the new reference. Thanks to reviewer for this suggestion. 

Reviewer 4 Report

I would recommend the revised MS "Investigation of Congo red toxicity towards different living organisms: A review"  for publication.

Author Response

Reply: Authors are thankful to reviewer for his/her positive response and acceptance of this manuscript. Again Thanks